

# Opinions regarding skin ageing in the elderly inhabitants of Bialystok, Poland

Mateusz Cybulski and Elzbieta Krajewska-Kulak

Department of Integrated Medical Care, Medical University of Bialystok, Bialystok, Poland

## ABSTRACT

Skin diseases constitute an essential health and aesthetic problem in the elderly. The aim of the study was to evaluate the knowledge of the elderly residents of public nursing homes and participants of the University of the Third Age in Bialystok, Poland surrounding the factors influencing skin ageing, the awareness of skin conditions in agening skin, and the impact of skin ageing on the volunteers. The study was performed from April to June 2015 in Bialystok, in two groups: among 100 public nursing home residents (PNH) and 100 members of University of the Third Age (U3A), (all over 60 years old). The study made use of a diagnostic survey conducted via a questionnaire prepared by the authors. Nearly half of those surveyed (42.5%; $n = 85$) sunbathed in the past, while 28.0% ($n = 56$) of those surveyed now take part in this type of leisure activity. More than half of respondents (53.0%; $n = 106$) protected their skin using special protective preparations. A majority of Bialystok inhabitants surveyed (80.5%; $n = 161$) noticed the features of skin ageing. They reported birthmarks, fungal infections and bedsores as the main skin problems of the old age. Nearly half (40%) of respondents assessed their knowledge as average and 26.0% as poor. The study showed some statistical differences in the knowledge and awareness between the residents of public nursing homes and the students of the University of the Third Age, e.g., the use of the Internet by the U3A group for finding out information. There is a desire to receive education in the field of the agening skin conditions/diseases among the elderly because their level of knowledge is relatively poor. Education of seniors in this area can increase their awareness of the basic principles of skin care and prevention marking of skin ageing. The benefits of greater knowledge of seniors about the conditions of agening skin can help reduce the medical burden and reduce the incidence on certain skin diseases. Furthermore, there is a need for educating of the younger population on the factors of skin ageing to prevent certain skin conditions as they become older. Seniors should be professionally educated by qualified specialists; for example, dermatologists or cosmeticians, so that the information they receive is in line with evidence-based medicine.

# INTRODUCTION

In the biological sense, the ageing of an organism is a process defined as a syndrome of progressive changes over time including, among others:

- decreased biological activity of cells;
- decreased regenerative processes;

Corresponding author
Mateusz Cybulski,
mateusz.cybulski@umb.edu.pl

- decreased immunity and response to an environmental stress;
- decreased genetic—dependent adaptation properties of an organism (*Zegarska & Wozniak, 2006*).

According to assumptions of the demographic projections performed by the *Central Statistical Office of Poland (2014)* up to the year 2050, a significant decrease in the population of children and adults and an increase in the number of the elderly is expected. According to the assumptions above, by 2050 people of at least 80 years in age will constitute 10.4% of the country's population, compared to 3.9% in 2013. Among 3.5 million of the Polish population aged 80 years and older in the year 2013, more than 59,000 will have lived to be 100 years old by 2050. In Poland, the longest-living inhabitants (especially women) are in the Podlasie province with its capital—Bialystok (*Central Statistical Office of Poland, 2014*).

The skin, similarly to all other body parts and tissues, is subjected to ageing due to intrinsic and extrinsic factors. Many analogies between ageing in skin and other organs can be found (*Kane, Ouslander & Abrass, 1998*). Wrinkles appearing on the face are one of the first signs of ageing (*Rexbye et al., 2006*).

Skin diseases constitute an essential health and aesthetic problem in the elderly. Many of these diseases are caused by long-term exposure to stressful situations and the impaired quality of the elderly person's life, including those caused by chronic diseases and smoking (*Daniell, 1971*; *Model, 1985*; *Kadunce et al., 1991*; *Ernster et al., 1995*), or overexposure to natural (*Leyden, 1990*; *Yaar, Eller & Gilchrest, 2002*), or artificial ultraviolet radiation (*Jayaprakasam et al., 2002*; *Shah & Coates, 2006*).

At present, smoking belongs to one of the two most serious extrinsic factors causing the most significant lesions in the skin (*Kennedy et al., 2003*), with the other being skin exposure to UV radiation (*Farage et al., 2008*). It has been proven in literature (*Leow & Maibach, 1998*) that smoking is a predictor of telangiectasia. Additionally, smoking cigarettes damages the skin by decreasing capillary blood flow in the skin, which results in a decrease in the volume of oxygen and nutrients in the dermal tissues. Other studies have shown that smokers have fewer fibres of collagen and elastin, which makes the skin loose and less elastic, facilitating wrinkles (*Leow & Maibach, 1998*). *Rexbye et al. (2006)* also proved that smoking was markedly associated with an increase in perceived age in men. It was established that smoking 20 cigarettes daily equalled an additional year in perceived age. For women, an additional year of skin ageing corresponded to smoking 20 cigarettes daily for 40 years (*Rexbye et al., 2006*). According to *Seddon et al. (1992)*, smoking 30 cigarettes daily at the age of 70 equalled about 14 years of skin ageing.

The aim of the study was to evaluate the knowledge of the elderly residents of public nursing homes and participants of the University of the Third Age in Bialystok, Poland surrounding the factors influencing skin ageing, the awareness of skin conditions in aged skin, and the impact of skin ageing on the volunteers. In reference to the aims of the study, we formulated the following hypothesis to test via questionnaire based study "knowledge and awareness of skin ageing is variable between elderly populations with different socioeconomic status."

## MATERIALS & METHODS

### Participants

The research was conducted in two groups—among 100 residents of Public Nursing Home (PNH) in 9 Swierkowa St. in Bialystok and among 100 participants of University of the Third Age (U3A) in Bialystok. A total of 200 people over 60 years old were included in the study.

The criteria of inclusion were defined as follows: age >60 years, respondents had to be inhabitants of Bialystok or the surrounding area, refer to the subject of the study and give written consent to participate in the study. An additional criterion of inclusion involving students of the U3A was the participation in lectures, because the study was conducted immediately before lectures. The criterion of exclusion was the incidence of dementia in the older person. Given the inclusion and exclusion criteria of the study, the research team sought to create a representative sample of the study groups. Therefore, it was decided to form two groups consisting of 100 respondents. The selection of groups was intentional and the choice of subjects in groups had a random character including the above criteria.

The authors decided to compare the results of research in the group of residents of PNH and participants of U3A, because it was assumed that there might be significant differences in the results due to socio-economic status between representatives of both groups.

### Measurements and procedurę

The study made use of a diagnostic survey conducted via a questionnaire prepared by the authors. It consisted of 35 questions, both open and closed, one or multiple choice. The questions concerned social-demographic characteristics (sex, marital status, place of living, education, financial status), respondents' knowledge of and use of basic preventative measures to avoid skin diseases (frequency of sunbathing and exposure to artificial UV light in solarium, protection of the skin against outer agents, applied cosmetics for skin care), knowledge of major skin problems in seniors (knowledge about factors affecting skin ageing, stigma of the senior age skin, the commonest skin problems among the elderly) and subjective assessment of the knowledge and its sources.

### Procedure and ethical considerations

The study was performed from April to June 2015. The research conforms with the Good Clinical Practice guidelines and the procedures followed were in accordance with the Helsinki Declaration of 1975, as revised in 2000 (concerning the ethical principles for the medical community and forbidding releasing the name of the patient, initials or the hospital evidence number) and with the ethical standards of the institutional committee on human experimentation (statute from the Bioethics Committee of the Medical University in Bialystok no. R-I-002/417/2014). All respondents gave written consent to fill out an anonymous questionnaire.

### Statistical analysis

All data obtained during the study were compiled using Microsoft Excel 2010. Statistical analysis was completed by applying the chi-squared test. Using the chi-square test, we

compared qualitative variables (groups) with quantitative variables (the type of cosmetics used, sources of information about dermatological diseases, etc.). The study meets all the conditions for the application of chi-square test, especially the minimum sample size and the independence of groups. Statistical hypotheses were verified at the $P \leq 0.05$ significance level, which means that it accepts the one error of the first kind in the 20 analyses. A confidence interval (CI) was calculated for each parameter using an $\alpha = 5\%$. Calculations were completed using STATISTICA Data Miner + QC PL program.

## RESULTS

### Baseline characteristics of the study population

In our study, women constituted 68.0% ($n = 136$), men—32.0% ($n = 64$). Among residents of PNH, there were 59 women and 41 men, while in members of U3A there were 77 women and 23 men. People over 70 years in age predominated (62.5%; $n = 125$), people aged from 61 to 70 years old ($n = 75$) constituted 37.5%. More than 40.0% of the surveyed (42.5%; $n = 85$) were widows/widowers, 30.0% were married, 15.0%—were singles and 12.0%—divorced. A significant majority of the respondents lived in the city—Bialystok (91.5%, $n = 183$), whereas only 5.5% ($n = 11$) in the rural area. A total of 45.0% of the surveyed ($n = 90$) defined their financial status as average, and 39.5% ($n = 79$) as good. Among PNH residents, 54.0% described their financial status as average, while 58.0% of U3A members considered as good. People with higher education (50.0%) and secondary education (44.0%) predominated in this group. Residents of PNH varied more with regard to the education level: 27.0% had vocational education, 26.0% had secondary education, 20.0% had primary education, 15.0% had higher education and 12.0% had technical education.

### Type of the skin and skin-associated problems of study participants

Almost one third of the total questioned (33.0%; $n = 66$) had a mixed type of the skin: in the PNH residents, this percentage equalled 35.0%, whereas in U3A members it was 31.0%. A slightly smaller percentage of people (25.5%, 51 people) had dry skin. An insignificant percentage of people had other types of skin. None of the respondents reported acne on their skin. A total of 21 people (10.5%) did not know the type of their skin.

More than 60.0% of all respondents (62.5%; $n = 125$), including 54.0% of PNH residents and 76.0% of U3A participants had no skin-associated problems in the past, whereas 28.5% of the respondents ($n = 57$) answered positively and 9.0% ($n = 18$) answered 'I don't know'.

### The frequency of sunbathing of the respondents

Almost half of the respondents from Bialystok (42.5%; $n = 85$) sunbathed in the past, but at present, they have stopped doing this. A total of 28.0% ($n = 56$) nowe take part in this leisure activity, whereas 29.5% had never improved the skin complexion in this way. Answers concerning indoor tanning were completely different. As many as 93.0% of the respondents ($n = 186$) had never used this type of services, 6.0% ($n = 12$) tanned indoors in the past, and only 1.0% ($n = 2$) at present. It should be highlighted that of the 14 people who had ever tanned indoors, 13 were the members of U3A.

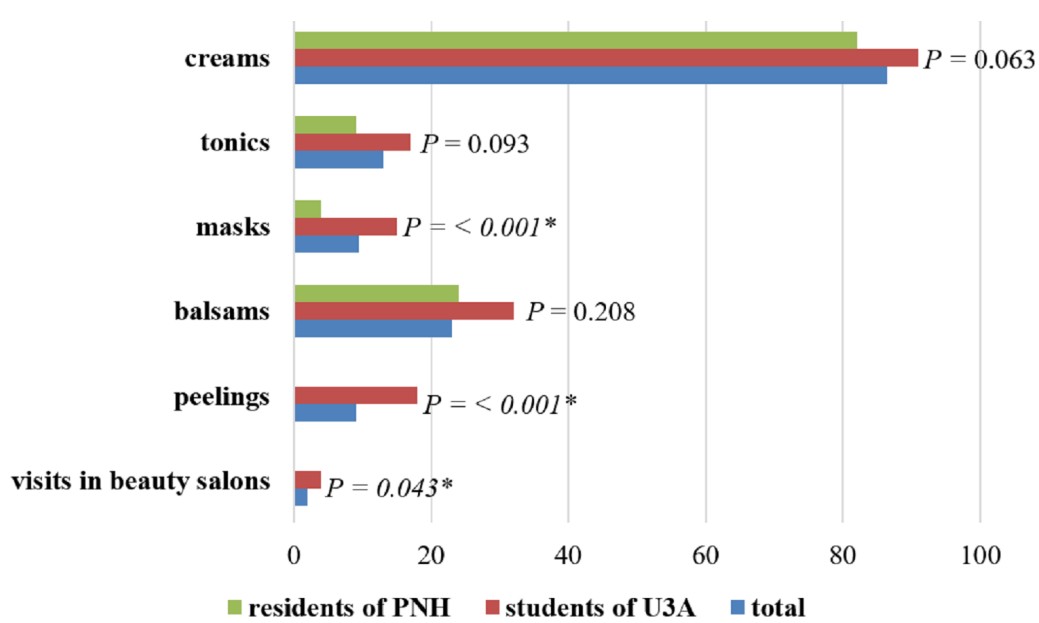

**Figure 1   Cosmetics for skin care used by respondents.** * Statistically significant difference ($P < 0.05$); PNH, Public Nursing Home; U3A, University of the Third Age.

### Use of special protective preparations by the respondents

Additionally, more than half of the respondents (53.0%; $n = 106$) protected their skin with special protective preparations. A quite high percentage of people (38.5%; $n = 77$) used no such preparations. Creams (86.5%; $n = 173$, $P = 0.063$) were most frequently used. Figure 1 shows the detailed data.

### The features of skin ageing of the study population

A majority of the respondents from Bialystok (81.5%; $n = 161$) showed signs of ageing skin. Only 9. 5% of the questioned observed no changes, and 10.0% were not able to state whether these changes occurred. General health conditions, smoking and stress were pointed to as factors influencing skin ageing (Fig. 2).

### Signs of ageing in the skin in the respondent's opinion

The most frequent signs of ageing in skin that were reported in the study are shown in Fig. 3. In the study, respondents most frequently pointed to wrinkles (80.5%), skin slackening (56.0%) and greying of hair (50.5%).

### Skin problems of the elderly

Respondents considered birthmarks, fungal infections and bedsores as main skin problems of the elderly. Detailed data, including the division into PNH residents and U3A participants and the value $P$ were presented in Fig. 4.

### Level of the knowledge on dermatological diseases

Those questioned did not have a high level of knowledge about dermatological diseases. More than 40.0% of all respondents (36.0% of PNH residents and 46.0% of U3A

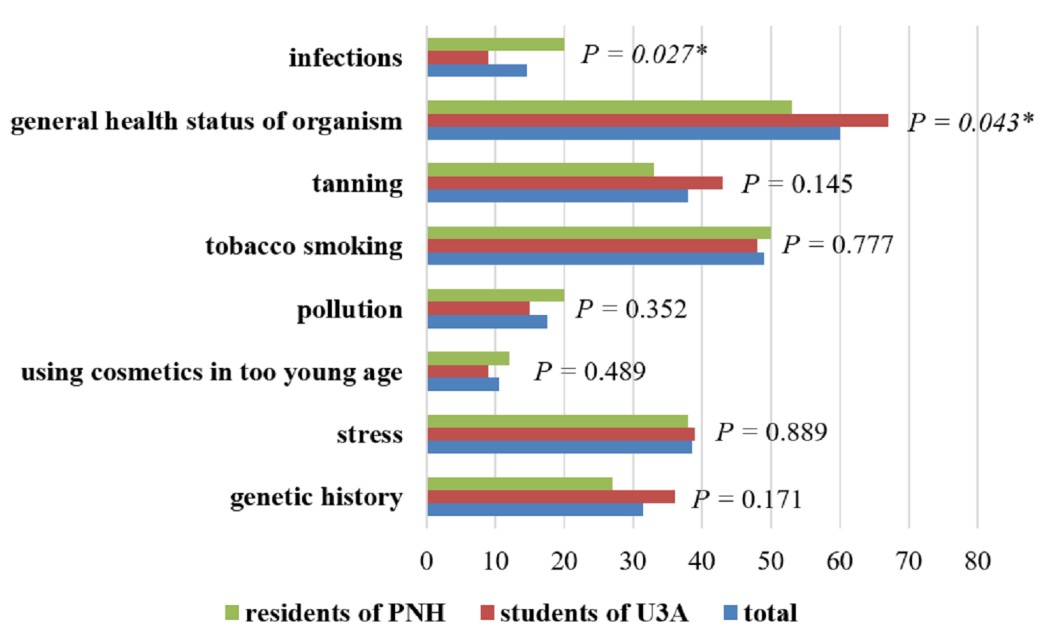

**Figure 2** **Factors influencing skin ageing in the opinion of respondents.** * Statistically significant difference ($P < 0.05$); PNH, Public Nursing Home; U3A, University of the Third Age.

participants) assessed their knowledge as average and 26.0% (35.0% of PNH residents and 17.0% of U3A participants) as poor. One person from U3A had very good knowledge and 24 people (12.0%) had good knowledge. U3A members also predominated in this case (16.0% of U3A members vs. 8.0% of PNH residents). The main sources of information about skin diseases of the elderly were newspapers and a physician (each 44.0%; $n = 88$). Reading newspapers was the most popular among U3A participants (50.0%), while the information obtained from a physician was more popular among PNH residents (55.0%). Other sources were shown in Fig. 5.

**The need for education about skin diseases among the elderly**
A total of 60.5% ($n = 121$) of respondents answered 'Yes' to the question 'Would you like to be educated with regard to skin diseases of the elderly?', while 8.5% ($n = 17$) answered 'No' and 31.0% ($n = 62$) had a problem to declare and chose the answer: 'It is difficult to say'. Analysis of 181 the group distribution of answers to the question revealed that 49.0% of PNH residents and as many as 72.0% of U3A members answered 'Yes', whereas 10.0% of PNH residents and 7.0% of U3A participants answered 'No'. Finally, 41.0% of PNH residents and 21.0% of U3A students answered 'I do not know'.

## DISCUSSION

Stasis dermatitis is one of the most frequent skin diseases in the seniors. This diseases is a cutaneous manifestation and marker of increased venous pessure of the lower extremities. It is a common condition affecting predominantly middle-aged to elderly individuals that usually presents as erythematous, slightly yellow to brown pigmented patches over the bilateral lower legs with or without conspicuous varicose veins. Most cases are caused by

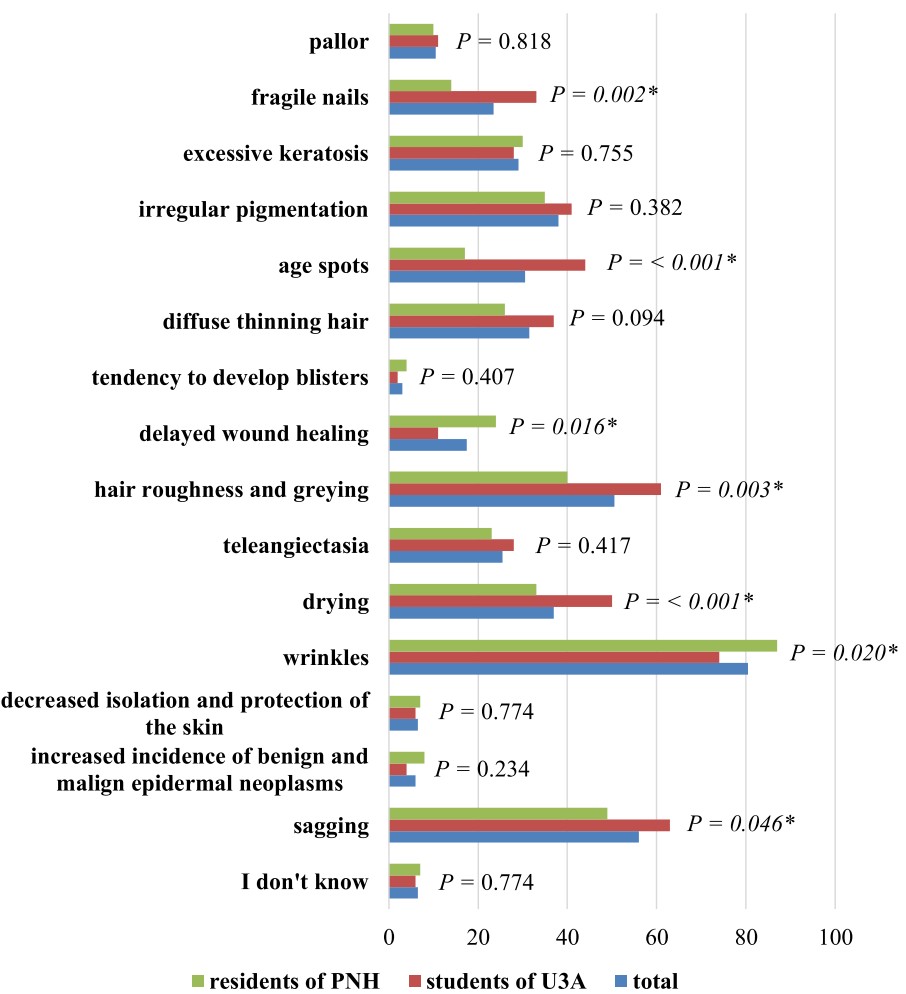

**Figure 3** **Typical stigma of ageing skin in the opinion of respondents.** * Statistically significant difference ($P < 0.05$); PNH, Public Nursing Home; U3A, University of the Third Age.

insufficient deep venous system valves preventing proper return of blood to the central circulation through the muscular pumping action of the lower legs. Venous valvular insufficiency can be caused by prior thrombophlebitis or congenital fragility. Obesity, and other causes of increased abdominal pressure can also lead to chronic venous insufficiency (*Weaver & Billings, 2009*). When left untreated or treated inadequately lesions may lead to ulceration of lower legs (*Thaipisuttikul, 1998*; *Baeke, 2000*). Lower legs ulceration was the most common cause of hospitalization among patients examined by *Pawlaczyk & Zukowska (2011)*. The second cause of hospitalization was eczema and next (more than 10%) was psoriasis (*Pawlaczyk & Zukowska, 2011*). Similarly, in the study of *Thapa et al. (2012)*, eczema (35.8%) was the most frequent form of dermatosis. Among geriatric patients of Taiwan, dermatitis, fungal skin infection and pruritic dermatosa were the most common skin problems (*Liao et al., 2001*). In Singaporean patients over 75 years old, eczema, dermatitis and epidermic keratoses were the most frequent skin diseases (*Yap, Siew & Goh, 1994*). In Germany, contact eczema, acne, and seborrheic dermatitis

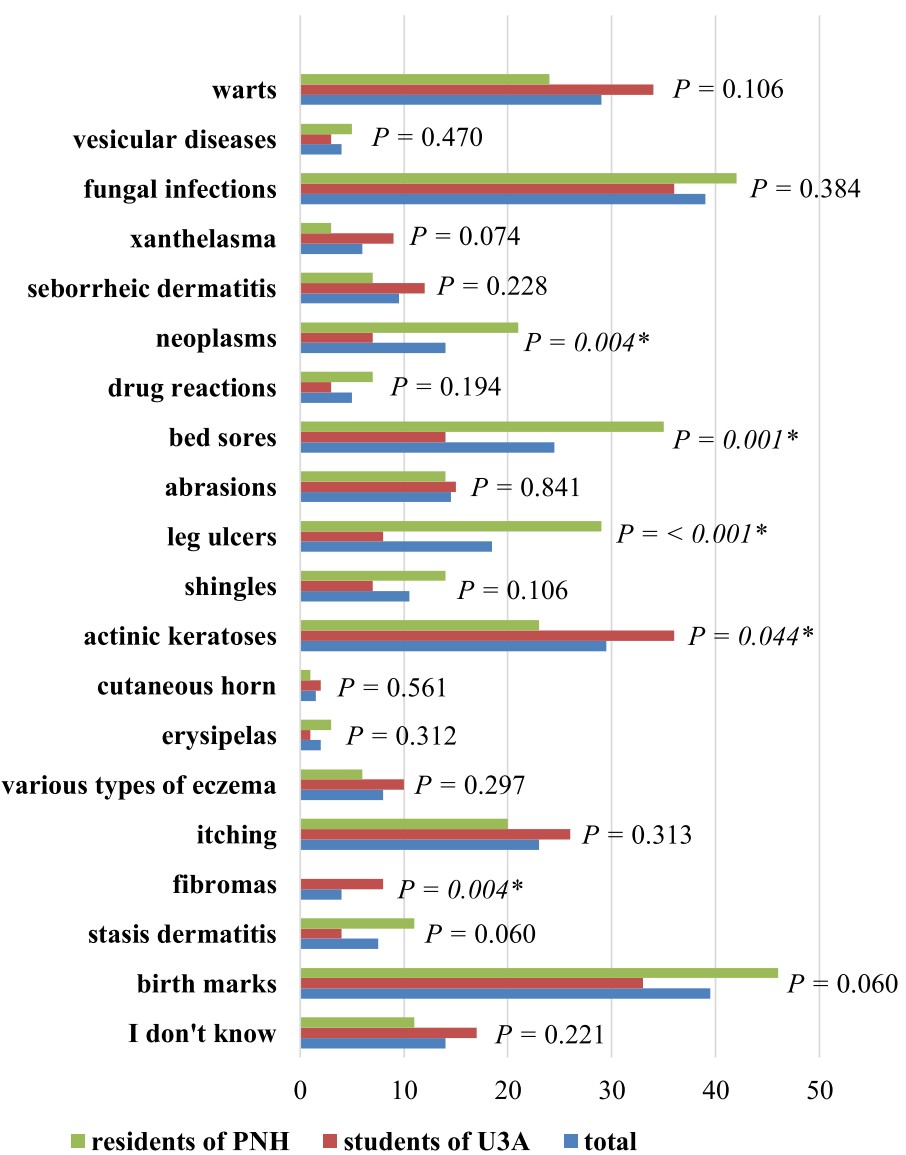

**Figure 4   The most frequent skin problems in the elderly in the opinion of respondents.** * Statistically significant difference ($P < 0.05$); PNH, Public Nursing Home; U3A, University of the Third Age.

were reported most frequently (*Schaefer et al., 2008*). In our study, respondents mentioned the most frequently birthmarks, fungal skin infections and decubitus ulcers as the most common skin problems of seniors. The respondents or members of their families might have had these types of skin problems.

U3A students pointed to fragile nails as the most common sign of ageing. This could be caused by exposure to moisture (e.g., washing the dishes), greater than among of PNH inhabitants. Among U3A participants, there were more women than among PNH residents. The available research suggests women are more vulnerable to fragility in nails than men. Age spots occur relatively frequently among U3A students than PNH residents. This could be due to greater exposure of U3A respondents to sunlight and ultraviolet radiation.
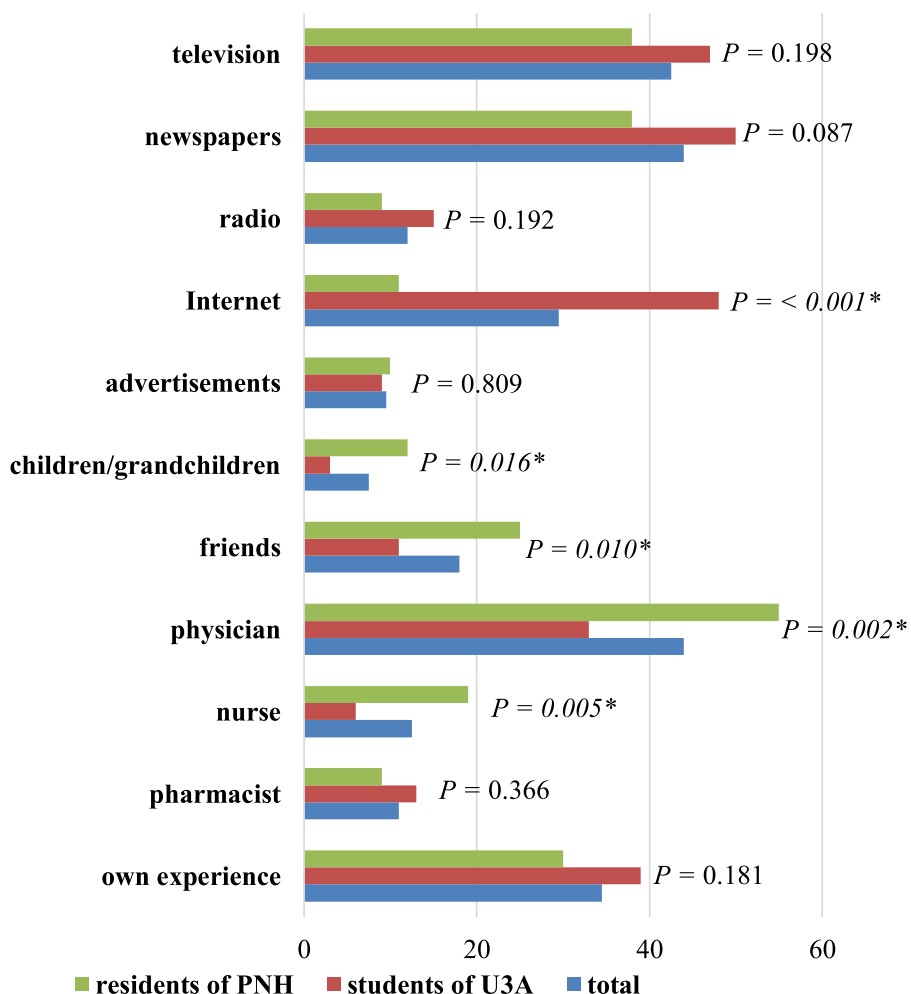

**television**   *P = 0.198*
**newspapers**   *P = 0.087*
**radio**   *P = 0.192*
**Internet**   *P = < 0.001\**
**advertisements**   *P = 0.809*
**children/grandchildren**   *P = 0.016\**
**friends**   *P = 0.010\**
**physician**   *P = 0.002\**
**nurse**   *P = 0.005\**
**pharmacist**   *P = 0.366*
**own experience**   *P = 0.181*

■ residents of PNH   ■ students of U3A   ■ total

**Figure 5** **Respondents' sources of information about skin diseases in seniors.** * Statistically significant difference ($P < 0.05$); PNH, Public Nursing Home; U3A, University of the Third Age.

Bedsores and leg ulcers are a greater problem for PNH. This could be due to a relatively large number of chronically ill patients. As a result, bedsores led to the formation of leg ulcers, which were also statistically more frequently indicated in this group of respondents.

The skin condition depends on the environmental conditions, like temperature and humidity. *McCallion & Li Wan Po (1993)* report that an increase in temperature by 7–8 °C doubles the loss of water evaporation, whereas the low temperature increases skin rigidity and decreases water loss by evaporation. In regard to the above information, the results of our study indicated that 53.0% of inhabitants of Bialystok questioned in the study protected the skin using special protective preparations against harmful atmospheric conditions like frost or wind.

In his study, *Rexbye et al. (2006)* proved that tobacco smoking was significantly associated with visual ageing. The same opinion about smoking cigarettes was expressed by the respondents in our study—they also considered tobacco smoking as the main factor determining skin ageing.

Solar radiation is responsible for 90% of visual skin ageing (*Martini, 2004*; *Sudel et al., 2005*). Solar radiation consists of three different types of radiation: UVC, UVB and UVA. UVC is to a great extent blocked by an ozone layer and has little influence on the skin (*Gilchrest, 1996*). UVB transfers exclusively to epidermis and is responsible for erythema (rash, redness) associated with sun burn (*Gilchrest, 1996*). UVA is characterised by a 1,000-fold higher radiation, causing sun burn at the same time. Additionally, it penetrates the dermis and is responsible for a majority of skin lesions, causing photoageing (*Gilchrest, 1996*). Moreover, UV radiation has been proven to have cancerogenic effects, as well (*Martini, 2004*). In our study, nearly three fourths of respondents have sunbathed at least once at their own choice, and they sunbathed more frequently in the past than in the present. It is reassuring that 93.0% of the surveyed had never tanned indoors, which has an even more harmful effect on the human health than natural solar radiation.

## CONCLUSIONS

1. The study showed some statistical differences in the knowledge and awareness between the residents of public nursing homes and the students of the University of the Third Age, e.g., the use of the Internet by the U3A group for finding out information.
2. There is a desire to receive education in the field of the old age skin conditions/diseases among the elderly because their level of knowledge is relatively poor. Education of seniors in this area can increase their awareness of the basic principles of skin care and prevention marking of skin ageing. The benefits of greater knowledge in seniors about the ageing skin can help reduce the medical burden and reduce the incidence on certain skin diseases. Furthermore, there is a need for educating younger population on the factors of skin ageing to prevent certain skin conditions as they become older age.
3. Seniors should be professionally educated by qualified specialists; for example, dermatologists or cosmeticians, so that the information they receive is in line with evidence-based medicine.

### Funding
The research was funded under grant no. 154-10956P of the Ministry of Science and Higher Education. The funders had no role in study design, data collection and analysis, decision to publish, or preparation of the manuscript.

### Grant Disclosures
The following grant information was disclosed by the authors:
Ministry of Science and Higher Education: 154-10956P.

### Competing Interests
The authors declare there are no competing interests.

## Author Contributions

- Mateusz Cybulski conceived and designed the experiments, performed the experiments, analyzed the data, wrote the paper, prepared figures and/or tables.
- Elzbieta Krajewska-Kulak conceived and designed the experiments, reviewed drafts of the paper.

## Human Ethics

The following information was supplied relating to ethical approvals (i.e., approving body and any reference numbers):

Bioethics Committee of the Medical University of Bialystok

Statute no. R-I-002/417/2014.

## Data Availability

The raw data has been supplied as Data S1.

## Supplemental Information

Supplemental information for this article can be found online at http://dx.doi.org/10.7717/peerj.2028#supplemental-information.

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
