# Peer review of "Opinions regarding skin ageing in the elderly inhabitants of Bialystok, Poland"

_PeerJ, doi:10.7717/peerj.2028_

## Round 0.1 · original submission · Major Revisions

· Academic Editor

Major Revisions

I agree with the comments of these reviewers, and recommend that you provide a much more robust justification for the experimental design (& its criteria) and its analysis in any resubmission. Define much more clearly the hypothesis under test (currently unclear), and why/how your test populations were selected. Provide a robust explanation for why you applied this statistical test to your data sets. There are very considerable English language issues here - please ensure that any resubmission is fully assessed by a native English speaker/writer.

·

Basic reporting

Overall the work and paper is poorly designed and reported. The relevance of the study to further our understanding of habits/skin condition in the aged is doubtful. The introduction requires additional information on why the study was done in this particular town in Poland and if the findings such as they were can be extrapolated to the general population, if at all.
The style, grammar and sometimes spellings (e.g. parlor for pallor) and use of terms that are either wrong or uncommon ( e.g. cancerogenic materials) in the report require a thorough critique and correction. Although the author probably has good English it is an insufficient standard for publication. An individual who has English as a first language should assist in achieving an more presentable report. The tables may not be the best means of showing the data. Exploring histogram presentations may be an option.

Experimental design

It is unclear as to why this particular population and location was chosen or its relevance to the rest of Poland or elsewhere. The inclusion and exclusion criteria and the format for entering a subject into the study was not explained.
It was said that 'The aim of the study was to evaluate the knowledge of the elderly residents of public nursing home and participants of University of the Third Age in Bialystok about the factors influencing skin aging.’ Although this was part of the investigation the main outcome appeared to be related to awareness of skin condition in aged skin and impact of skin ageing on the ageing volunteers.

Validity of the findings

The Conclusions in the report stated:
'As a result of conducted research the following conclusions have been formulated:
1. There is a need for education of seniors about skin diseases in the elderly age, because their level of knowledge is relatively low.
2. Seniors should be educated professionally by qualified specialists, for example dermatologists, cosmeticians.
3. The information the elderly obtain should be established in accordance with evidence –based medicine, not on colourful magazines for women.'

It was not discussed why the elderly needed to know about skin diseases or what was the effect of ignorance of this subject or what benefit would this education bring?
It would seem more appropriate to educate the younger population on the factors of skin ageing to prevent certain skin conditions as people get older.
Conclusions 2 and 3 did not come directly from information in the study but are opinions expressed by the author

Additional comments

The study report requires a complete rewrite. Ask the question what was I trying to find out, what did I find out, what is the significance of what I found out. how will this benefit the health of the elderly in the future. None of this is very clear from the current report.

Reviewer 2 ·

Basic reporting

Language and style changes are captured in the annotated pdf. Please review suggested changes, which would help make the manuscript clearer.
In table 3 please change "parlor" to "pallor".

Experimental design

No comments

Validity of the findings

Some numbers will need to be checked for accuracy.

Additional comments

It would be nice to see the questionnaire as seen by the respondents to better understand the questions they were asked, possibly as an appendix or supplementary material. Some medical terms in the questionnaire may have been difficult to understand for a person of a non-medical background and may have influenced their responses to the questions. Was an explanation given to the terminology?

Annotated reviews are not available for download in order to protect the identity of reviewers who chose to remain anonymous.

·

Basic reporting

The authors could describe the potential advantages and disadvantages of this type of study design and mention any similar types of work exploring this topic.

The authors should set out their initial hypothesis based on the context explained in the introduction. The authors could consider whether they expect any relationships to exist in the information sought.

The results are described in text and tables. This is duplication and not required. Tables are long and cumbersome. Authors should consider an alternative format for presentation of results. For example, it is not necessary to include 'n' and % given the total number of respondents is known. Graphs should be considered also some analysis of whether responses are clustered or more biased to one of the two groups. The authors might like to consider whether some parts of the discussion should be moved into the introduction; especially where only prior work is discussed. Ideally the discussion should relate study findings to previous work, where possible.

It may be worth considering writing a results and discussion section to facilitate a more critical analysis of the findings.

Experimental design

The authors should more clearly explain why the two study populations were chosen, taking into consideration socio-economic and other factors that might influence the study findings.

The authors could explain the value of using the Chi squared test of significance in this type of study and the meaning and implications behind a significant result. The * in tables at a significant finding refers to a baseline, this needs to be explained.

Validity of the findings

The study appears to conclude that elderly residents are less well informed than would be ideal about skin ageing but this is just one area of questioning; what is the evidence when considering groups of the questions that this is the case?

The authors might like to consider what benefits such education might lead to and authors could make some comment on any differences between the two populations investigated and why these differences exist.

Additional comments

This study examines in some detail the levels of knowledge of elderly people about factors important in skin ageing. This type of study is not often done, so is useful. However, the authors have an opportunity to improve the manuscript by adding some critical analysis of findings against a stated hypothesis and to more clearly explore whether and why responses are different in the two groups studied.

A further draft should be reviewed by a person with excellent written English, however, I do realise some direct translation issues in the questions might be an issue.

The authors amply cite the literature and could make more comments or suggestions on what their data adds to the general understanding of skin ageing. Can any popular myths be disproved?

Have the authors thought to use further/different analysis methods to see whether certain questionnaire responses are clustered, or tried to present the data using graphs?

---

## Round 0.2 · Major Revisions

· Academic Editor

Major Revisions

Unfortunately, I could not see where you have updated the manuscript - rather I see one version with the entire text deleted (i.e. in red cross-out marking) followed by a text version with no highlighted changes. Please resubmit your revision showing where and how the manuscript has been revised according to the reviewers' comments.

Also, it would be useful to provide the questionnaire translated into English, as well as a justification for the statistical tests(s) used.

---

## Round 0.3 · Minor Revisions

· Academic Editor

Minor Revisions

The reviewers acknowledge the improvements made to this revision, but also suggest ways to improve it further in order to bring this manuscript closer to a standard needed for publication. I agree with their assessment, and strongly recommend the authors closely look at these comments, especially reviewer 2, and strive to address them. Where this is not possible, please provide an explanation to support why this is so.

·

Basic reporting

The rewrite has created a better document for publication. The aims are clearer and the overall structure has improved. This is particularly noticeable in the description of the aims and how the study was conducted. The discussions and relevance is clearer although the conclusions tend to be generic and not specific. The relevance of health improvement was not addressed. The graphic presentation of the data helps in understanding the outcome of the study.

Experimental design

This is a relatively simple study where the important aspects are the inclusion and exclusion criteria of the subjects and questionnaire design and execution.
These were adequately explained.

Validity of the findings

The findings were clearly explained and to a degree put into context with other studies in several countries.
The authors have not yet made it clear as to the benefits of greater knowledge of seniors about ageing skin condition. Apart from the benefit of improved knowledge would this help reduce the medical burden or reduce incidence on certain skin conditions? If the authors could address this then it would round off the conclusions and findings.

·

Basic reporting

The manuscript has been much improved. The written English is much clearer and the authors addressed the comments of the reviewers.

Experimental design

The design of the study is now well explained.

Validity of the findings

There is a disappointing lack of interpretation and discussion of the results and the conclusions still do not reflect the results obtained.

Additional comments

The authors have made a lot of very good improvements. The graphs are very good, but care should be taken to accurately explain the statistical comparators in the legend.

The English sometimes is not quite reading well. I suggest some changes below:-
Page 4
In reference to the aim of the study was formulated the following research hypothesis „There are significant differences in knowledge and awareness of skin aging in the elderly people constituting study groups”.

In reference to the aims of the study we formulated the following hypothesis to test via questionnaire based study “knowledge and awareness of skin ageing is variable between elderly populations with different socioeconomic status”

Page 4
….respondents’ use of basic prophylaxis of skin diseases (frequency of sunbathing and exposure to artificial UV light in solarium, protection of the skin against outer agents, applied cosmetics for skin care),

….respondents’ knowledge of and use of basic preventative measures to avoid skin diseases (frequency of sunbathing and exposure to artificial UV light in solarium, protection of the skin against outer agents, applied cosmetics for skin care),……

Page 5
Using the chi-square test was compared qualitative variables (groups) with quantitative variables (the type used cosmetics, sources of information about dermatological diseases, etc.).

Using the chi-square test, we compared qualitative variables (groups) with quantitative variables (the type of cosmetics used, sources of information about dermatological diseases, etc.).

Page 7
Signs of the skin in the respondent’s opinion
The most frequent signs of the skin reported in the society were shown in Fig. 3.

Signs of ageing in the skin in the respondent’s opinion
The most frequent signs of ageing in skin that were reported in the study are shown in Fig. 3.

Those questioned had no high opinion about their knowledge on dermatological diseases.

Those questioned did not have a high level of knowledge about dermatological diseases.


Discussion
Stasis dermatitis is one of the most frequent skin diseases in the seniors.

Could you explain what stasis dermatitis is for the non-expert reader?

In regard to the above information, the results of our study proved that 53.0% of inhabitants of Bialystok questioned

In regard to the above information, the results of our study indicated that 53.0% of inhabitants of Bialystok questioned

(you did not have direct proof as this was a questionnaire study)


Figures
What is the baseline to which you refer in all of the figure legends?


Conclusion 1.
The figures suggest some statistical differences between the two groups? So how you state overall no difference?
Can you highlight some of the more interesting ones; e.g.
the use of the internet by the U3A group for finding out information?
Why are fragile nails and age spots more reported in the UEA group as a sign of ageing???
Bedsores and leg ulcers are a greater problem for the PNH – why might that be??

---

## Round 0.4 · accepted · Accept

· Academic Editor

Accept

The revisions have improved this paper further. I still have some concerns with the quality of the English used in the writing of the manuscript. I would therefore like to request that you continue to work with the Journal staff to improve this further at production phase.